# Investigating the effectiveness of structured mindfulness sessions in mitigating burnout among final-year dental students: A mixed-methods analysis

Aliya Islam[1]*, Javeria Rehman[2], Sumbul Mujeeb[3], Rubab Javed[4], Unaiza Hashmi[5]
Tazeen Saeed Ali[6]

**1** Faculty of Health Professional Education, Baqai Medical University, Karachi, Sindh, Pakistan,
**2** Department for Educational Development and Pathology & Laboratory Medicine, Medical College,
The Aga Khan University,Karachi, Sindh, Pakistan, **3** Department of Psychology, Sir Syed University
of Engineering and Technology, Karachi, Sindh, Pakistan, **4** Office of Research, Innovation, and
Commercialization (ORIC), Khyber Medical University, Peshawar, Khyber Pakhtunkhwa, Pakistan,
**5** Institute of Public Health, University of Lahore, Lahore, Punjab, Pakistan, **6** Department of Community
Health Sciences, School of Nursing and Midwifery,The Aga Khan University, Karachi, Sindh, Pakistan

* dr_aliya48@hotmail.com

## Abstract

### Background

The transition from dental student to practicing dentist is often accompanied by high level of stress and risk of burnout. This shift from academic to professional roles can overwhelm students and compromise their well-being. Addressing burnout early is essential to maintain academic performance, resilience, and quality patient care. Mindfulness interventions have shown increasing empirical support in reducing stress and burnout. This mixed-methods study evaluated the effectiveness of mindfulness-based life skill-building sessions in alleviating burnout among final-year dental students at Baqai Dental College, Karachi, Pakistan.

### Method

A sequential mixed-method study was conducted at Baqai Dental College in Karachi, Pakistan, involving sixty-nine final-year dental students. The intervention comprised two mindfulness sessions. Participants completed pre- and post-intervention questionnaires that included demographic information and the Copenhagen Burnout Inventory–Student Survey (CBI-SS), a validated 25-item tool assessing burnout across four domains: personal, study-related, colleague-related, and teacher-related. Post-intervention questionnaires were completed immediately after the sessions. Qualitative data were obtained through focus group discussions with stakeholders to explore perceptions of burnout and assess the relevance and feasibility of mindfulness sessions. Quantitative data were analyzed using paired t-tests, while qualitative

**Data availability statement:** The datasets generated and analyzed during the current study contain potentially identifiable information about human participants and cannot be publicly shared due to institutional privacy and ethical restrictions. Data access is governed by the Ethics Review Board of Baqai Medical University, Karachi, Pakistan. Qualified researchers may request access to the anonymized data by contacting the Ethics Review Board at registrar.secretariat@baqai.edu.pk. All relevant summary data supporting the findings of this study are included within the manuscript.

**Funding:** The author(s) received no specific funding for this work.

**Competing interests:** The authors have declared that no competing interests exist.

data were examined through content analysis and triangulated with literature and documentation.

## Result

Diverse emotions and experiences were revealed through qualitative analysis, underscoring the substantial influence of exhaustion on personal, academic, and professional domains. Furthermore, the mean exhaustion scores were significantly different between the pre- and post-intervention periods, with a p-value of 0.001. Students conveyed satisfaction and a desire for more meaningful sessions that bolster life-building skills.

## Conclusion

The study findings demonstrate a decrease in burnout rates following the intervention, indicating the necessity for interventions to mitigate academic stress. Qualitative findings underscore the significance of a student support centre and self-care initiatives.

## Introduction

Burnout syndrome (BO) is a psychological condition characterized by a sustained reaction to persistent emotional and interpersonal pressures, especially within the professional and academic environment [1]. The term "burnout" was first introduced by Herbert Freudenberger in the 1970s to describe physical and emotional exhaustion among young volunteers and was later conceptualized by Christina Maslach and Susan E. Jackson as a multidimensional syndrome involving emotional exhaustion, depersonalization, and reduced personal accomplishment [2]. The World Health Organization (WHO) classifies burnout as an occupational phenomenon in the ICD-11, defining it as a syndrome arising from chronic workplace stress that has not been successfully managed [3].

Burnout syndrome is an increasing concern among dental students, characterized by emotional exhaustion, cynicism, and professional inefficacy [4], often stemming from the demanding nature of this discipline [5,6]. Studies indicate that up to 53.8% of dental students experience significant stress [7], which may lead to burnout syndrome [8]. In Pakistan, over 85% of medical and dental students report experiencing burnout syndrome [9]. Academic pressure, demanding curricula, frequent examinations, and fear of failure contribute to this condition, adversely affecting students' mental health and academic performance [6] and potentially compromising the quality of future patient care [10]. Nonetheless, the understanding of this issue remains unrecognized among students and educators [11]. Effectively managing burnout is crucial to avert mental health issues, such as intentional self-harm and suicide [12,13]. Institutions must enhance consciousness to elevate student well-being and, consequently, the quality of patient care [11,14]. Although the issue is acknowledged,

a deficiency persists in evidence-based interventions specifically designed for dentistry curricula [5,15].In Pakistan, approximately 55 recognized dental colleges are offering a four-year Bachelor of Dental Surgery (BDS) program, followed by a one-year house job (internship) under the regulations of the Pakistan Medical & Dental Council (PMDC) [16]. The curriculum is predominantly discipline-based and follows a traditional model, in which subjects are taught in separate blocks, including didactic lectures, pre-clinical training, and clinical rotations [17]. At Baqai Dental College, students follow a rigorous academic and clinical schedule [7], with assessments that include written examinations, practical assessments, viva voce, and OSCE/OSPE formats. This structure, characterized by a heavy workload, long hours, and high-stakes assessments, can contribute significantly to student stress and burnout [18,19].

Innovative ways are required to incorporate mental well-being programs into academic and clinical environments [10], with special emphasis on the function of mindfulness in promoting resilience and improving student well-being [20].

Implementing interventions at both the institutional and community levels is crucial for enhancing the dental education system [21,22]. Although existing data underscores the frequency of burnout among dentistry students, the development and execution of evidence-based interventions remain insufficient.

A potentially useful technique to mitigate burnout among dental students is mindfulness-based Life Skill Building sessions. The World Health Organization characterizes mindfulness as the capacity to remain present and respond without judgment, thereby providing pupils with vital life skills to manage life's challenges adeptly [23]. This practice entails methodical cognitive training that enhances self-awareness and promotes effective self-regulation for addressing daily issues [24]. Although mindfulness is useful in alleviating stress and improving emotional regulation in numerous situations [24].Its application in dental education and the perceived efficacy among students remain insufficiently examined and necessitate additional investigation. This establishes a significant deficiency in comprehending how to utilize mindfulness as an effective strategy to alleviate burnout among dentistry students, a domain predominantly neglected in current studies.

This research seeks to provide an innovative coping approach, "mindfulness LSB sessions," to mitigate burnout among dental students. This study aimed to examine the effect of mindfulness Life Skill Building workshops on alleviating burnout and enhancing the overall well-being of final-year dental students.

## Methodology

### Ethics approval and consent to participate

Ethical approval for this study was obtained from the Ethics Review Committee of Aga Khan University (Letter No: 2023-8517-25732) and the Ethical Review Board of Baqai Dental College (Reference No: BDC/ERB/2023/043). All methods were carried out in accordance with relevant guidelines and regulations, including the Declaration of Helsinki. Informed consent was obtained from all participants before data collection.

### Research framework

A mixed-methods study design was employed, incorporating both qualitative and quantitative components to thoroughly investigate the research topic. The research commenced with a sequential exploratory framework during the qualitative phase, subsequently transitioning to a quasi-experimental design in the quantitative phase to assess the effectiveness of mindfulness LSB sessions in alleviating burnout among final-year dental students.

### Study context and population

The research was carried out at Baqai Dental College, a private institution located in Karachi, Pakistan, and involved participants who were final-year dental students.

## Duration of the study

The research was carried out over a duration of nine months, with participant recruitment starting on **09/08/2023** and concluding on **06/04/2024**, following ethical approval from the Ethical Review Committees of Aga Khan University (No. 2023-8517-25732) and Baqai Dental College (ERB Ref. No. BDC/ERB/2023/043).

## Sample size and sampling methodology

**Qualitative segment.** The qualitative component aimed to gather expert perspectives to perspectives to inform the development of the Mindfulness-Based Life Skill Building (LSB) curriculum. The 12 participants were stakeholders, not students, and were not part of the burnout assessment cohort. Purposive sampling was used for the selection process, which included faculty members responsible for overseeing final-year students (Head of Department, senior lecturers), a student psychologist, a counsellor, and alumni, chosen for their expertise and direct engagement in providing academic and psychological support. Three focus group discussions were conducted, each comprising four participants. Faculty members who satisfied the inclusion criteria yet opted out of participation or failed to attend the interview were excluded from the study.

**Quantitative arm.** A comprehensive sampling method was employed to recruit all 69 final-year Bachelor of Dental Surgery (BDS) students at Baqai Dental College. All students attended both mindfulness LSB sessions and completed the study assessments, resulting in a 100%participation rate. Although exclusion criteria were defined (students unwilling to participate or absent from more than one session), no participants met these criteria; therefore, no students were excluded from the analysis.

## Data collection method

The overall process of data collection is illustrated in **Fig 1**.

**Formative phase: development of the mindfulness-based Life Skill Building (LSB) curriculum.** The formative phase focused on the contextual development of the Mindfulness Life Skill Building sessions through qualitative assessment. A preliminary draft of the curriculum was developed by the researcher following a thorough review of international literature on mindfulness-based educational interventions, particularly those addressing stress, burnout, and emotional well-being among healthcare students. To ensure contextual relevance and responsiveness to the participants' needs, three focus group discussions (FGDs) were conducted. These discussions explored key academic stressors, perceived contributors to burnout, and participant expectations regarding the structure and content of the proposed sessions. Emergent themes such as emotional exhaustion, inadequate coping strategies, and the need for enhanced self-regulation directly informed the refinement of the curriculum. The final version was reviewed by a subject matter

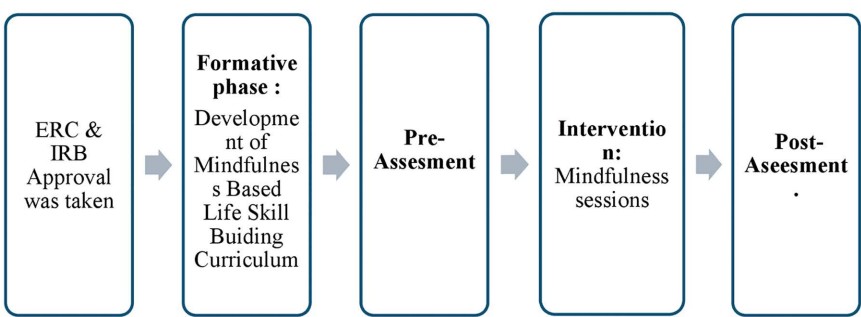

**Fig 1. Flowchart of the phases of data collection.**

expert to ensure theoretical rigor and practical relevance. This expert input further guided the design of session activities, with a particular focus on core mindfulness practices and essential life skills, including self-awareness, interpersonal communication, and the development of self-esteem.

**Pre-assessment phase.** To establish a comprehensive baseline of burnout and inform the implementation of mindfulness-based Life Skill Building sessions, a mixed-methods approach was used. Qualitative data were gathered through three focus group discussions (FGDs) conducted with alumni and faculty members who had firsthand experience with the academic and emotional challenges faced by dental students. These 60-minute discussions followed a semi-structured guide and explored key issues such as academic stress, institutional pressures, coping strategies, and expectations regarding the structure and delivery of mindfulness-based interventions. While these FGDs primarily informed the development of the curriculum, they also offered valuable contextual understanding of the stressors commonly encountered by students, supporting the rationale for the intervention.

Subsequently, a quantitative assessment was conducted using the Copenhagen Burnout Inventory Student Survey [25], a validated tool with strong psychometric properties (Cronbach's alpha = 0.929). The CBI-SS was chosen for its reliability, validity, and practicality in assessing burnout among students. It is a free, accessible tool that measures multiple dimensions, personal, studies-related, colleague-related, and teacher-related, providing a comprehensive understanding of student burnout and enabling identification of specific predictors for targeted interventions. Approval for use of the adapted Thai version was obtained from the original author and co-author [25,26]. Before administration, the tool was pilot tested among third-year BDS students, who confirmed its clarity and comprehensibility. The CBI-SS comprises 25 items across four domains: personal burnout (6 items), study-related burnout (7items), colleague-related burnout (6 items), and teacher-related burnout (6 items). Each item is rated on a five-point Likert scale, reflecting frequency: Never (0%), Rarely (25%), Sometimes (50%), Frequently (75%), and Always (100%), with higher scores indicating higher levels of burnout. All items are scored in the same direction, and no items are reverse-scored. Total scores for each subscale were calculated by summing the responses to the respective items, with higher subscale scores representing higher levels of burnout. The questionnaire, along with demographic data (age, gender, academic year), was administered in a four-hour session supported by the Department of Medical Education. A 20-minute briefing session explained the study's purpose, procedures, risks, benefits, and voluntary nature. Informed consent was obtained in writing from all participants, and those who declined were excluded from the study. This combined pre-assessment approach provided a robust baseline to evaluate the impact of the intervention.

**Intervention: mindfulness sessions.** Final-year BDS students participated in two mindfulness sessions, each lasting two hours, facilitated by a clinical psychologist and a co-facilitator certified in neuro-linguistic programming and transformational mind technologies. Both sessions were held on the same day for logistical reasons, with a 45-minute break in between to accommodate the availability of facilitators and ensure the comfort and engagement of participants.

The first session focused on burnout awareness and stress management, introducing students to the concept of mindfulness and its relevance in academic life. The second session emphasized the practical application of mindfulness-based Life Skill Building strategies, addressing areas such as self-awareness, interpersonal communication, and self-esteem.

Each session integrated guided mindfulness exercises, experiential activities, journaling prompts, and open group discussions to encourage personal reflection and peer engagement. The content was strategically aligned with the burnout-related challenges identified in the pre-assessment phase, including emotional exhaustion, low resilience, and ineffective coping strategies. Visual aids and structured scripts were used to ensure delivery consistency, and facilitators actively engaged students through reflective dialogue. Attendance was recorded, and student reflections were collected to monitor engagement. The overall aim was to provide students with both a theoretical framework and practical tools to manage stress and enhance emotional well-being.

**Post-assessment.** Following the mindfulness sessions, a post-test utilizing the same CBI-SS questionnaire was administered to assess the effectiveness of the intervention. To guarantee precision and reduce bias, burnout scores were

computed from both pre- and post-assessments. Students exhibiting significant burnout according to Kristensen's criteria [27] were scheduled for referral to the psychiatry department for comprehensive evaluation and support, with counseling to be administered in accordance with institutional policy.

To further assess the perceived impact of the intervention, four open-ended questions were added at the end of the CBI-SS. The inquiries facilitated students in articulating their experiences regarding the mindfulness sessions, yielding significant insights characterized by a range of responses that underscored both favourable evaluations and potential areas for enhancement.

**Analysis of data and strategy for qualitative investigation.** Data for the focus group discussions were gathered in confidential sessions facilitated by a trained moderator. Consent from participants was secured for audio recording, and observational notes were documented to record nonverbal signals. The researcher collaborated with a professional transcriber to execute a verbatim transcription. Additionally, member checking was implemented to verify the accuracy of the transcripts. Field notes underwent validation through the analysis of participant data, adhering to Creswell's framework. This process entailed the condensation of transcripts into significant statements, which were subsequently refined into distinct categories and themes. Inductive coding was employed to discern patterns, which were subsequently refined into categories and themes. The integration of literature, recordings, and field notes facilitated the validation of the findings.

**Analysis of data for quantitative investigation.** For the quantitative analysis, pre- and post-test data were examined utilizing SPSS version 22 to compute mean burnout scores and subscale scores. The Shapiro-Wilk test confirmed the presence of normality, and the histogram exhibited no signs of skewness. The burnout scores were reported using the mean and standard deviation. A paired t-test was performed for inferential analysis, establishing statistical significance at a 95% confidence interval and a p-value of ≤ 0.05.

## Result

### Qualitative analysis

A total of three focus groups, each comprising 12 participants, were organized to investigate essential concepts related to burnout. Focus group discussions 1 and 2 comprised eight alumni (five females, three males), all of whom were house officers with a consistent institutional affiliation of five years (Mean±SD: 5±0.00). FGD 3 consisted of four faculty members: one male associate professor with 7 years of experience, two female assistant professors with 3 and 5 years of experience, respectively, and one female senior instructor with 5 years of experience, resulting in an overall mean tenure of 5±1.63 years. The discussions revealed three primary themes **(see Fig 2)**: *manifestations, contributing factors*, and *potential solutions*, providing a comprehensive understanding of the elements influencing student burnout.

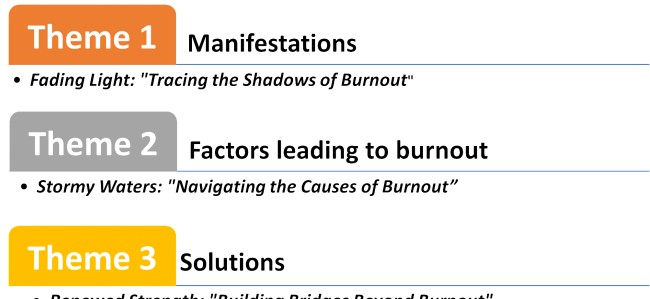

**Theme 1** Manifestations
- *Fading Light: "Tracing the Shadows of Burnout"*

**Theme 2** Factors leading to burnout
- *Stormy Waters: "Navigating the Causes of Burnout"*

**Theme 3** Solutions
- *Renewed Strength: "Building Bridges Beyond Burnout"*

**Fig 2. Thematic analysis showing three primary themes identified from the qualitative discussions.**

**Theme 1 – fading light: tracing the shadows of burnout (Manifestations).** Burnout is portrayed as a **"diminishing light,"** a fading glow that casts heavy **shadows** across the emotional, psychological, and physical well-being of dental students. These shadows reflect the multifaceted toll that burnout takes, often dimming motivation, performance, and overall wellness (**Fig 3**).

**Sub-theme 1: Emotional and Psychological Impact**: Burnout doesn't just exhaust students; it overwhelms them emotionally. Participants described a pervasive sense of **helplessness**, **self-doubt**, and **detachment** from both personal and academic responsibilities.

*Behavioral Changes:* Burnout seeped into daily behavior, manifesting as **irritability**, **anger**, and **overthinking**.

*"I struggled to sleep due to constant overthinking, which led to disrupted eating habits and a cranky mood. I used to lose my temper and get into arguments easily."* (FGD1-P3)

It also strained important relationships, even in simple moments:

*"Burnout has not just affected me personally but strained my relationship with my family, friends, and teachers. For instance, I would respond with anger even when my mom asked something nicely."* (FGD2-P4)

*Demotivation and Neglecting Personal Life:* The emotional toll often spiraled into **demotivation** and a retreat from everyday life.

*"Eventually, there comes a point when you don't feel like working; you don't feel like doing your daily activities."* (FGD2-P3)

For some, burnout drained the joy from achievements:

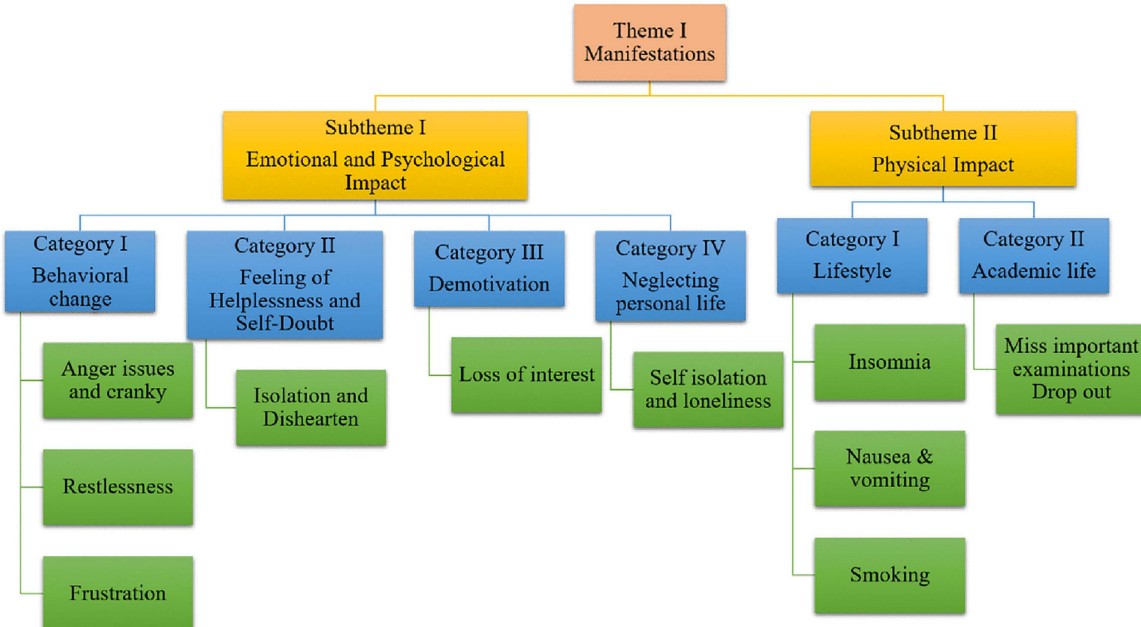

**Fig 3. Subthemes and categories for theme 1, illustrating the emotional, psychological, and physical impact of burnout on dental students.**

*"It feels like there is no desire to do anything, you're just working to survive. Burnout drains you of happiness and the sense of achievement you used to feel."* (FGD3-P1)

**Sub-theme 2: Physical Impact**: Beyond emotional and behavioral effects, burnout also left **visible marks on students' physical health,** from disrupted sleep to changes in eating habits.

*"I lost my sleep & appetite. People ask me why I'm so skinny, but they don't understand. Sometimes, parents remain unaware of the burden colleges place on us."* (FGD1-P1)

For some, the toll was so severe it led to dropping out of academic programs:

*"Physically, the dropout rate increased."* (FGD3-P2)

**Theme II – stormy waters: navigating the causes of burnout (factors leading to burnout).** This theme conceptualizes burnout as a challenging and unpredictable journey in which dental students face continuous academic pressures, personal responsibilities, and societal expectations. Without adequate coping mechanisms and institutional support, these stressors can accumulate and lead to emotional and psychological exhaustion. The theme underscores the importance of fostering resilience and providing a structured support system to help students manage these challenges effectively (**Fig 4**).

This theme is divided into three interconnected sub-themes:

**Sub-theme 1: Academic Stress**: At the forefront of burnout lies the intense **academic pressure** that students face. Participants highlighted that **packed schedules and insufficient breaks** drained their energy and left little room for recuperation or self-care.

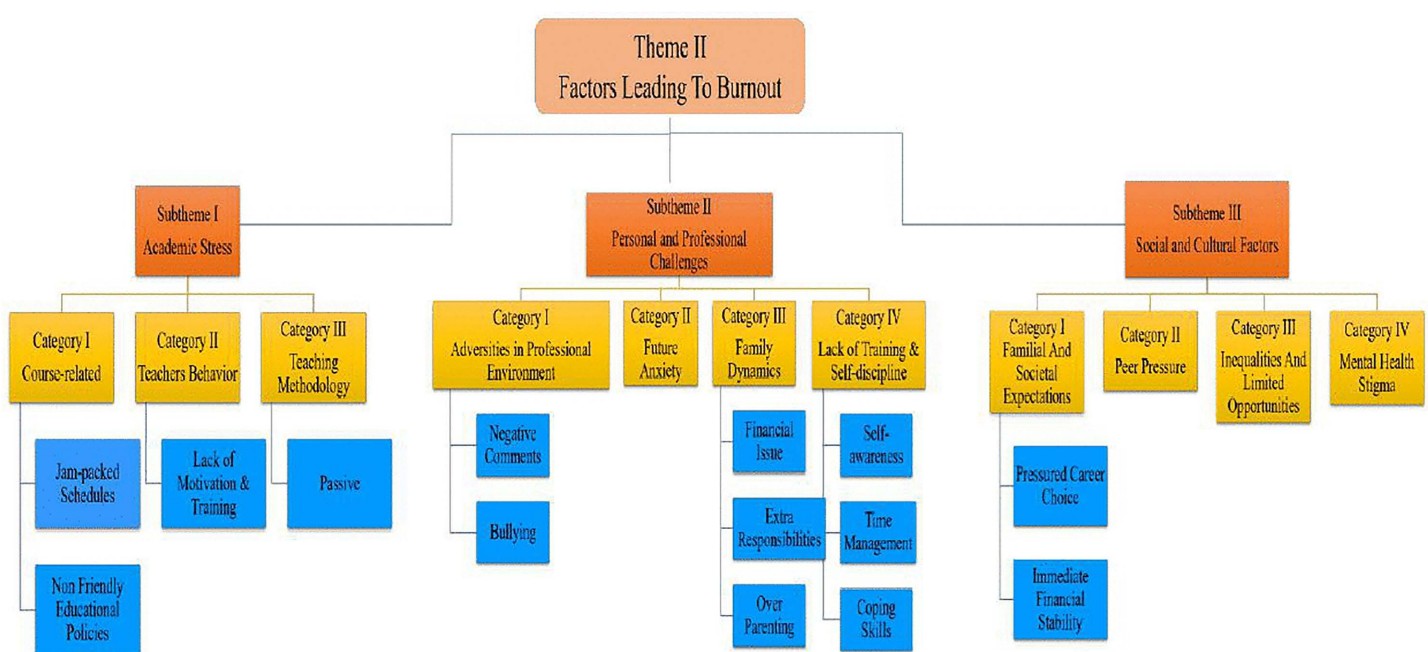

**Fig 4. Subthemes and categories associated with factors contributing to burnout among dental students.**

*"The timetable lacks sufficient breaks between classes, leaving students fatigued and with limited time for other activities."* (FGD3-P3, P4)

These sentiments call attention to the urgent need for more **balanced timetables** and student-centered academic planning.

Equally impactful is the **student–teacher dynamic**. A lack of pedagogical support amplified feelings of isolation and discouragement:

*"Uncooperative behavior of teachers where they are unwilling to assist students in understanding the topic and expect the students to understand without further explanations."* (FGD2-P4)

This highlights the importance of **empathetic and responsive teaching approaches** that go beyond rote delivery to foster understanding and emotional safety.

**Sub-theme 2: Personal and Professional Challenges**: Burnout is also shaped by a range of **personal and professional demands**, especially for students juggling multiple roles. Some, particularly female participants, described the weight of **academic and household responsibilities** colliding.

*"University pressure adds to household responsibilities, creating a dual burden."* (FGD2-P1)

Students also spoke of **bullying**, **harsh feedback**, and the lack of emotional support, all of which compounded their stress. Moreover, **financial constraints**, **family expectations**, and **limited autonomy** added layers to the burden.

These experiences emphasize the need to foster **self-awareness**, **coping mechanisms**, and **time management skills** critical tools for students navigating such dual roles and high expectations.

**Sub-theme 3: Social and Cultural Factors**: Burnout among dental students does not occur in isolation; it is often **shaped by the social and cultural environments** they inhabit. Family expectations, societal ideals, and peer comparisons all influence students' academic journeys.

Many felt pressure to pursue dentistry not out of passion but as a **symbol of family prestige**, leading to inner conflict and stress.

In parallel, **cultural stigma around mental health** has created barriers to seeking help, further deepening feelings of isolation.

*"Parents often don't understand the toll it takes. The pressure to keep up academically while not being able to talk about stress makes it harder."* (paraphrased theme insight)

This sub-theme underscores the urgent need to **normalize conversations around mental health**, reduce stigma, and develop culturally sensitive support systems.

**Theme III – Illuminating the Way: Strategies to Reduce Burnout (Solutions).** This theme captures student-generated strategies to mitigate burnout through both **counselling-based** and **institutional-level interventions**. These proposed solutions serve as "guiding lights" illuminating a path toward more supportive, responsive, and student-centered academic environments **(Fig 5)**.

This theme is organized into two major sub-themes:

**Sub-theme 1: Counselling and Mental Health Support**: Students strongly demanded structured counselling services that provide both emotional and career guidance. Regular **one-on-one sessions with trained facilitators** were seen as essential for managing academic stress and personal challenges.

*"Career counselling holds equal importance."* (FGD2-P1)

*"There is no career counselling in our society, and it plays an important role in emotional well-being."* (FGD1-P1)

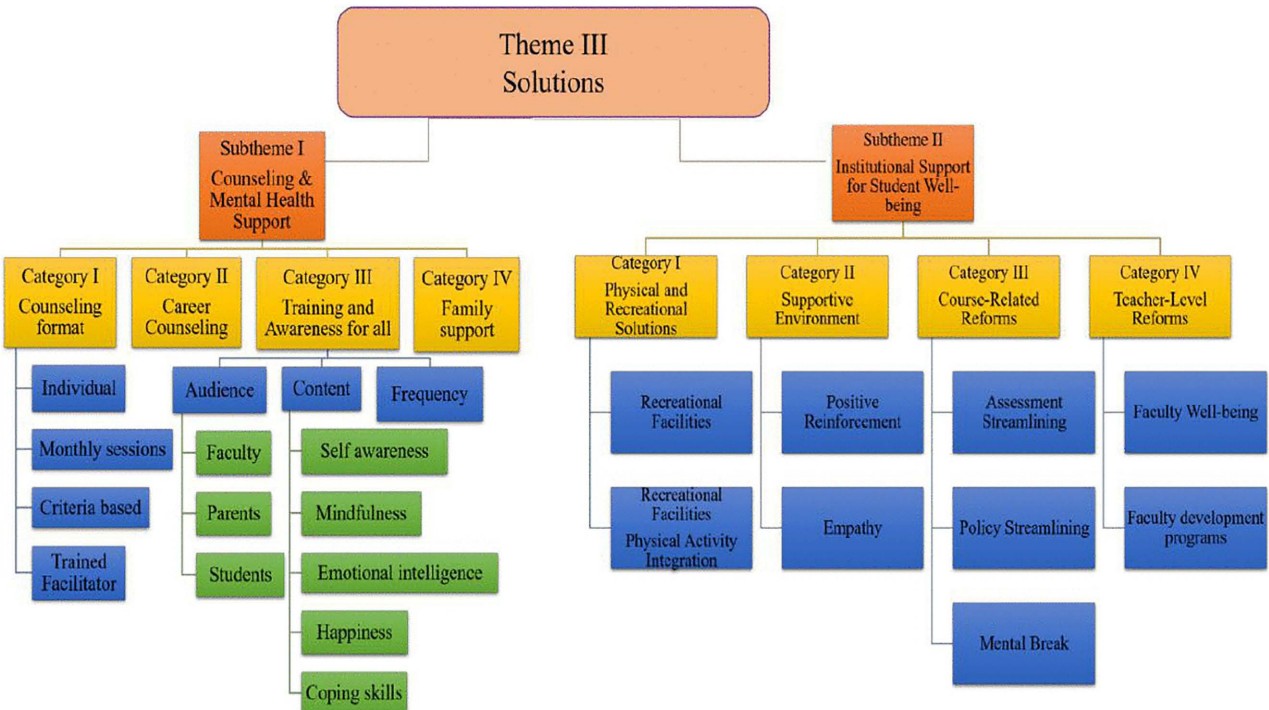

**Fig 5. Subthemes and categories representing strategies to reduce burnout among dental students.**

Beyond counselling, participants emphasized the need for skill-building in **self-awareness, emotional intelligence**, and **interpersonal communication**. Many students struggle to identify or process their emotions, leaving them vulnerable to the effects of stress:

> *"Emotional intelligence is lacking; they don't know how to identify emotions. If they don't know what they are going through, how will they address it, process it, and cope with it?"* (FGD3-P1)

To address this, students proposed **mindfulness activities** and **communication-focused workshops** aimed at building resilience, fostering peer support, and promoting emotional regulation.

**Sub-theme 2: Institutional Support for Student Well-being**: Participants stressed that **institutional reforms** are just as important as individual support. Their suggestions fell into four key domains:

**1. Physical and Recreational Solutions:**

Students called for **dedicated spaces for recreation and structured physical activities** to help decompress from academic strain.

> *"There should be a time for physical exercise and a place for recreational activities."* (FGD3-P3)

**2. Supportive Academic Environment:**

They emphasized the value of a **culture grounded in empathy and positive reinforcement** where educators not only teach, but also inspire, support, and motivate.

### 3. Course-Related Reforms:

Concrete curriculum changes were suggested, including revisiting course loads, modifying assessments, and integrating **scheduled mental breaks** to reduce academic overload.

*"Redesign the curriculum."* (FGD2-P2)

### 4. Teacher-Level Reforms:

Faculty development was identified as a cornerstone for sustainable change. Participants recommended training teachers in mental health sensitivity and student engagement techniques.

*"Teachers should also be counselled and trained for taking sessions."* (FGD3-P4)

## Quantitative analysis

This research assessed the effects of mindfulness sessions on burnout levels in final-year dental students, utilizing the Copenhagen Burnout Inventory Student Survey (CBI-SS) within a quasi-experimental framework. A total of sixty-nine students, comprising 32 males and 37 females, with a mean age of 22.5 years (± 1.016), participated in the CBI-SS assessment both prior to and following the intervention. The Shapiro-Wilk test validated the presence of a normal distribution within the data set. The dependent variable scores were reported using mean and standard deviation, revealing a statistically significant difference in mean burnout scores before and after the intervention ($p < 0.001$), as illustrated in Table 1.

Following the analysis of pre- and post-scores of burnout across four domains, statistically significant reductions were observed in two domains. As shown in Table 2, personal burnout decreased to a post intervention mean of 13.69, and study-related burnout decreased to 15.90. In contrast, reductions in colleague-related burnout (12.30 to 11.69) and teacher-related burnout (11.07 to 10.38) were observed but were not statistically significant. A paired t-test was used to compare the differences between pre- and post-intervention burnout scores across all domains. Reductions were observed in two domains, while decreases in the other two domains were not statistically significant.

## As illustrated in Table 3, A Statistical Analysis of the Effectiveness of Interventions on Burnout Domains

The effectiveness of the intervention on various burnout domains among students was assessed using the Copenhagen Burnout Inventory – Student Survey in a paired sample t-test. The findings provide compelling evidence of positive change, while also elucidating the complex landscape of student well-being.

● **Personal Burnout: A Significant Increase in Emotional Exhaustion**

Personal burnout was the domain in which the most remarkable transformation occurred. The intervention resulted in a statistically significant decrease in pre- and post-intervention scores (M = 1.62, SD = 4.40; t(69) = 3.05, p = 0.003), indicating that students experienced a meaningful decrease in emotional and physical exhaustion. This discovery implies that the

**Table 1. Paired sample statistics of mean burnout scores with P-Value.**

| Burnout measures | Mean (Pre-intervention) | Mean (Post-intervention) | Mean Difference (Pre-Post) | Standard Deviation | Standard Error of the Mean. | 95% Confidence interval (Lower) | 95% Confidence interval (Upper) | t | P-Value |
|---|---|---|---|---|---|---|---|---|---|
| Total Burnout Score | 62.67 | 54.47 | 8.19 | 17.67 | 2.14 | 3.92 | 12.47 | 3.82 | < 0.001 |

**Table 2. Paired sample statistics of all domains.**

Paired Samples Statistics
Paired Samples Statistics

| Domains | | Mean | N | Standard. Deviation | Standard. Error of Mean |
|---|---|---|---|---|---|
| Personal Burnout | Pre-intervention | 15.30 | 69 | 3.57 | .42 |
| | Post-intervention | **13.68** | 69 | 3.13 | .37 |
| Study-related burnout | Pre-intervention | 17.77 | 69 | 4.40 | .53 |
| | Post-intervention | **15.80** | 69 | 3.33 | .40 |
| Colleague-Related Burnout | Pre-intervention | 12.29 | 69 | 5.29 | .63 |
| | Post-intervention | 11.6932 | 69 | 4.25 | .51 |
| Teacher-Related Burnout | Pre-intervention | 11.06 | 68 | 4.80 | .58 |
| | Post-intervention | 10.37 | 68 | 4.58 | .55 |

**Table 3. Paired sample test of all domains.**

Paired T test on all the domains of Copenhagen burnout inventory student survey

Paired Sample Test

| Burnout Domain | Mean difference (Pre-Post) | Standard Deviation | Standard Error of Mean | 95% Confidence interval | | t | P-Value |
|---|---|---|---|---|---|---|---|
| | | | | Lower | Upper | | |
| Personal Burnout | 1.61 | 4.39 | 0.52 | 0.55 | 2.67 | 3.05 | 0.003 |
| Study-related burnout | 1.87 | 5.32 | 0.64 | 0.59 | 3.15 | 2.92 | 0.005 |
| Colleague-Related Burnout | 0.60 | 5.42 | 0.65 | −0.69 | 1.90 | 0.92 | 0.358 |
| Teacher-related burnout | 0.68 | 5.79 | 0.70 | −0.71 | 2.09 | 0.98 | 0.330 |

intervention not only reached the students but also resonated with them on a personal level, potentially providing strategies or perspectives that facilitated their reconnection with their internal energy reserves. The intervention assisted in the rekindling of the students' "inner spark," thereby reducing emotional fatigue and reestablishing a sense of individual control.

● **Burnout Related to Studying: Academic Pressure Begins to Reduce**

Changes in Study-Related Burnout were equally promising. The post-intervention mean was markedly lower (M = 1.88, SD = 5.32; t(69) = 2.92, p = 0.005), indicating a statistically significant reduction in academic exhaustion. This suggests students may have acquired improved coping strategies, time management skills, or experienced a greater sense of support and visibility enabling them to manage study more effectively. This indicates that students were better able to manage their workloads and cope with academic demands.

● **Colleagues-Related Burnout: No Significant Change**

Conversely, there was no statistically significant change in Colleague-Related Burnout and other types of burnout (M = 0.60, SD = 5.42; t(69) = 0.92, p = 0.358). Although the mean difference was positive, the high variability and wide confidence interval indicate that improvement in peer-related burnout was inconsistent. This suggests that fatigue associated with social or collegial relationships necessitates a more focused or prolonged intervention, or potentially structural changes to group interactions.

● **Teacher-Related Burnout: No Statistically Significant Change**

Lastly, there was no significant change in Teacher-Related Burnout (M = 0.69, SD = 5.79; t(69) = 0.98, p = 0.330). This implies that the stress experienced by students as a result of faculty interactions and expectations remained essentially

unchanged. It is feasible that the institutional and hierarchical nature of faculty–student relational dynamics necessitates additional systemic reforms or direct faculty involvement in the fatigue mitigation process. Improvements in student outcomes may depend on concurrent development of faculty communication and support strategies.

**Reliability**: The internal reliability of the Copenhagen Burnout Inventory Student Survey used in this study was assessed using SPSS, yielding a coefficient of 0.916.

### Students' feedback regarding the effectiveness of mindfulness life skill building sessions

The students provided positive feedback regarding the mindfulness sessions, with many expressing a desire for more sessions. They found these sessions beneficial for enhancing motivation, promoting mental relaxation, and improving life management skills. Most students reported significant reductions in burnout, stress, and anxiety, although a few described the sessions as moderately helpful. There was strong support for regularly integrating these sessions into the curriculum, with suggestions to emphasize academic stress management and consider shorter session durations.

### Data integration

Triangulation in this study involved the integration of qualitative insights and quantitative findings to enhance methodological rigor and deepen understanding of the multifaceted nature of burnout among students. Both data strands converged to reveal the **emotional, behavioral, and physical toll** of burnout, particularly stemming from academic overload, social pressures, and insufficient time for self-care.

Quantitative data, analysed through paired t-tests, demonstrated **statistically significant reductions in personal and study-related burnout** following mindfulness-based interventions. These results were echoed in the qualitative narratives, where participants described feeling more emotionally grounded, better equipped to manage stress, and more connected to their internal states after engaging in the intervention.

Participants strongly advocated for institutional support mechanisms such as mental health counselling, mindfulness workshops, and faculty training to create a more empathetic academic environment **(Fig 6)**. This **convergence of findings** validated the intervention's positive impact, confirmed the relevance of student-recommended strategies, and supported the **rejection of the null hypothesis**. Ultimately, the triangulated results provided a comprehensive, credible, and contextually rich understanding of how burnout can be mitigated through a combined focus on individual resilience and systemic reform.

## Discussion

The primary objective of this study was to evaluate the necessity, feasibility, content, and efficacy of a mindfulness-based intervention in reducing burnout among final-year dental students in Pakistan. The study provides compelling evidence that burnout is not only prevalent among dental students but also profoundly ingrained in their academic, emotional, and social realities, by employing the CBI-SS in a sequential mixed-methods design.

Before the intervention, high levels of exhaustion were observed in all domains: personal, study-related, teacher-related, and colleague-related, as per quantitative analysis. Despite the substantial decreases in personal and study-related exhaustion, stressors related to teachers and colleagues continued to persist. These findings suggest the presence of structural and cultural factors that may require approaches extending beyond individual-level coping strategies.Overall, the results indicate that mindfulness-based interventions were associated with improvements in emotional resilience and reducing psychological distress,although these effects were not uniform across all burnout domains.

The quantitative findings were supported by the qualitative data, which provided additional context and depth. Students were forthcoming about the emotional, physical, and behavioral experiences associated with academic life, including reduced motivation,strained interpersonal relationships, and disturbances in appetite and sleep. Qualiataive results also highlighted challenges to emotional awareness, self-regulation, and perceived self-efficacy skills that are not consistently

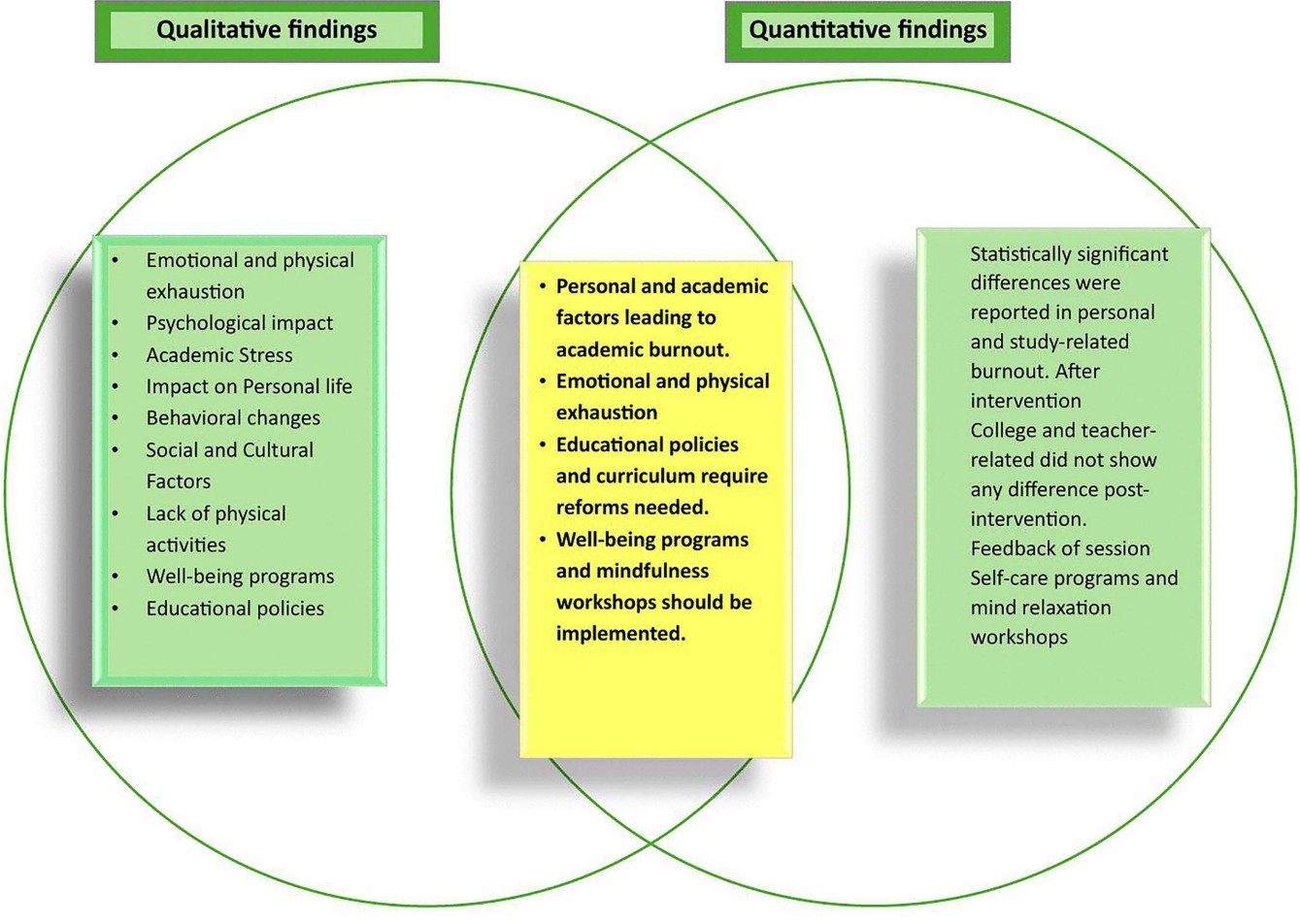

**Fig 6. Venn diagram illustrating the interrelationships among the main themes.**

emphasized within current curricula.These factors were associated with emotional exhaustion and disengagement from both academic and personal activities,findings that are consistent with global research, including work by João Maroco and studies conducted in Pakistan and Brazil [28,29].

The qualitative phase identified a recurring theme in which exhaustion was described as both normalized and stigmatized. Students reported that burnout was seldom discussed and was often dismissed as a sign of frailty [30]. Participants described an academic culture that emphasizes endurance while discouraging the expression of vulnerability [11]. This pattern,compounded by unmet expectations, academic workload, and perfectionistic tendencies,suggest a largely unrecognized challenge within the healthcare education [1,11].

The findings indicate that the mindfulness intervention was supported by the triangulation of qualitative and quantitative data. The post-intervention reduction in CBI-SS scores, from a pre-intervention mean of 62.67 to 54.47 was statistically significant and was further contextualized by the narratives of the students.Participants described improved emotionally awareness, more composure during clinical interactions, and an increased ability to manage academic stress. Collectively, these findings provide evidence supporting the role of structured mindfulness interventions in enhancing emotional regulation among dental students.

The lack of statistically significant improvement in burnout related to teachers and peers should not be disregarded. These findings suggest that supporting student emotional competence alone may be insufficient without corresponding attention to faculty behaviors and institutional practices. The persistence of burnout may therefore reflect the need for complementary interventions targeting educators and broader systemic factors. Consistent with studies conducted in Hungary and Malaysia, the present findings suggest that variations in fatigue may be linked to learning environments, social support systems, and faculty-student relationships [31,32].

A key strength of this study is its methodological approach. It is among the first studies in Pakistan, one of the few internationally to evaluate the impact of mindfulness in undergraduate dental education using the CBI-SS pre- and post-intervention design. This quasi-experimental design, in contrast to previous cross-sectional or correlation studies, enables both statistical and thematic validation and captures change over time.

In addition, the study incorporates qualitative data from focus groups and open-ended responses to provide insights into students' lived experiences. The findings indicate that exhaustion may be influenced by social and institutional factors, including gender norms, familial expectations, academic hierarchies, and institutional practices, rather than solely reflecting individual-level clinical outcomes.

### Limitations

The study was limited to a single institution and city, indicating that future research could enhance its scope by adopting a multicentric approach involving both private and public sector institutes across various provinces.

The study exclusively focused on final-year dental students from a single institution, without including participants from other academic years within dentistry and other health professions. This scope may limit the generalizability of the findings beyond this specific group. Additionally, since the study was conducted only once on a single batch for a short period over one year, it only assessed short-term effects and cannot predict the long-term impact of this intervention based on these findings.

A potential limitation of the study was the relatively less number of mindfulness sessions, limited to only 2 sessions on the same day, which may have influenced the burnout outcomes. Increasing the number of sessions might enhance the intervention's impact and lead to better future outcomes for the participants. Therefore, future research could consider incorporating additional mindfulness sessions to further explore the intervention's potential effectiveness.

Additionally, as burnout was assessed through self-reported measures, the responses may have been influenced by individual perceptions or social desirability bias. This subjectivity could have led some students to under- or over-report their experiences, potentially affecting the precision of the findings.

Thus, the findings reflect the effectiveness of the intervention in this context, rather than its efficacy under controlled conditions.

### Conclusion and future directions

This investigation presents a compelling argument in favour of systemic reform. It illustrates that the integration of mindfulness-based sessions in a thoughtful manner can substantially mitigate personal and academic burnout. Nevertheless, in order to effect meaningful, enduring change, institutions must transcend isolated interventions. In addition to training faculty to establish inclusive, compassionate learning environments, they must incorporate modules on resilience-building, self-care, and emotional intelligence into the curriculum.

Workshops on communication, emotional regulation, and time management, in addition to weekly mindfulness sessions, should be implemented as an integral component of healthcare education. Institutional counseling centers must be fortified and destigmatized. National policymakers should prioritize student well-being as a fundamental outcome of academic excellence.

The mental health of pupils is not an optional consideration in a high-stakes, high-stress field such as dentistry; it is a necessity. The present investigation provides a replicable, evidence-based model for the implementation of mindfulness as the initial stage in the development of a health education system that is more emotionally intelligent, responsive, and humane.

## Acknowledgments

I would like to thank my supervisor, Dr. Tazeen, and co-supervisor, Dr. Javeria, for their valuable guidance and support. My sincere gratitude goes to the final-year dental students at Baqai Dental College for their participation. I also appreciate the resources and support provided by Baqai Dental College. Finally, I am grateful to my family and friends for their constant encouragement.

## Author contributions

**Conceptualization:** Aliya Islam.

**Data curation:** Rubab Javed, Unaiza Hashmi.

**Methodology:** Aliya Islam.

**Project administration:** Sumbul Mujeeb.

**Supervision:** Javeria Rehman, Tazeen Saeed Ali.

**Validation:** Tazeen Saeed Ali.

**Writing – original draft:** Aliya Islam.

**Writing – review & editing:** Javeria Rehman, Tazeen Saeed Ali.

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
