## [Decision Letter · Decision Letter 0]

25 Sep 2025

Dear Dr. Yasir,

Thank you for submitting your manuscript to PLOS ONE. After careful consideration, we feel that it has merit but does not fully meet PLOS ONE’s publication criteria as it currently stands. Therefore, we invite you to submit a revised version of the manuscript that addresses the points raised during the review process.

We look forward to receiving your revised manuscript.

Kind regards,

Jenny Wilkinson, PhD

Academic Editor

PLOS ONE

Journal Requirements:

3. In this instance it seems there may be acceptable restrictions in place that prevent the public sharing of your minimal data. However, in line with our goal of ensuring long-term data availability to all interested researchers, PLOS’ Data Policy states that authors cannot be the sole named individuals responsible for ensuring data access (http://journals.plos.org/plosone/s/data-availability#loc-acceptable-data-sharing-methods).

**Additional Editor Comments:**

This is an interesting study and both reviewers have highlighted areas that can be expanded or further explained to strengthen the manuscript.

Reviewers' comments:

Reviewer's Responses to Questions

**Comments to the Author**

1. Is the manuscript technically sound, and do the data support the conclusions?

Reviewer #1: Yes

Reviewer #2: No

2. Has the statistical analysis been performed appropriately and rigorously?

Reviewer #1: Yes

Reviewer #2: Yes

3. Have the authors made all data underlying the findings in their manuscript fully available?

Reviewer #1: No

Reviewer #2: Yes

4. Is the manuscript presented in an intelligible fashion and written in standard English?

Reviewer #1: Yes

Reviewer #2: Yes

Reviewer #1: A very interesting research article which thoroughly explores an important topic, presenting results in a clear detailed way. Relates well to and builds on previous research, whilst also identifying key areas for further research. Please see my comments and suggestions below.

No concerns about ethics.

Re data availability - the authors have stated it is not possible to publicly share all data due to its confidential nature. However details of access have been provided.

Abstract and introduction

- good summary of the research though a bit wordy

- identifies gaps in literature which should be filled and how this study relates to previous research

- consider including a brief explanation about how the dentistry course in Pakistan works - How many years is it? How long are the days? How are they assessed/taught?, this could provide a bit more context for international readers who are unaware

Methods

- explanation of participant selection is a bit unclear, were the 12 participants staff selected to lead the focus groups? Or were these 12 participants the remaining students from the original 69 once you had eliminated those who did not attend sessions etc

- were the individuals in the initial focus groups members of staff, or the students themselves - ie: who is helping to develop the LSB curriculum?

- the two above points become more clear in the “pre-assessment phase” section, I think it is the “qualitative segment” section which causes confusion - consider specifying at the beginning of this paragraph that the 12 participants were selected for the focus group rather than for measuring burnout

- why was the CBI-SS tool selected over other tools to measure burnout?

- how much time passed between carrying out the first and second mindfulness sessions? Consider exploring in the discussion whether this may have had an effect on burnout levels

Results

- what was the overall participation rate? It is mentioned that some students were excluded (eg: because they did not attend both mindfulness sessions), overall what % of final year dental students participated in the full study? - if low, mention the limitations of this in the discussion

- some of the themes seemed wordy and overly descriptive, I didn’t feel that describing dental students as “weary sailors” helped to contribute towards the value of this research

- similar to the above point - line 538 in the discussion - “provided a rich texture” feels overly descriptive and not in-keeping with the nature of the paper

Discussion

- an interesting discussion which summarises and explores the findings well

- are there any changes which could be made to the focus group to have a more significant effect on colleague and teacher related burnout? Or can this only be achieved by changes made to the faculty?

- little mention of the limitations of this study, this is a key area to consider, especially as burnout and stress can be so subjective and vary massively between individuals (eg: this was only carried out on students in a specific year group at a specific dental school, did the time of year influence the burnout levels of students, ie were the higher burnout levels recorded close to exam periods?)

- burnout is quite a sensitive topic, consider whether students may have been under- or over- reporting their experiences and how this may pose a limitation

Conclusion

- succinct conclusion which also clearly identifies key areas for future research and action, as well as conveying the importance off this research topic

Statistical analysis

- I do not have sufficient expertise to report on the adequacy of the statistical analysis

Figures and tables

- I felt that the figures helped to map the themes out in an easy to understand way, and summarises well the many factors contributing to burnout in dental students

Reviewer #2: I appreciate the opportunity to review this manuscript. The authors have presented findings from their study on the efficacy of mindfulness practices as essential competencies in mitigating burnout among final-year dental students. Unfortunately, I am unable to recommend this manuscript for publication due to the following reasons:

* Burnout (BO) needs to be framed as a syndrome in the manuscript. According to the World Health Organization and based on the seminal work of Dr. C. Maslach and Dr. S.E. Jackson, burnout syndrome results from prolonged exposure to chronic stress. Providing detailed background information on burnout syndrome is important as it relates to the manuscript’s title and key words the authors chose.

* Based on how the literature defines “efficacy” this study design did not appear to meet standards to measure the efficacy of the mindfulness-based therapies that were examined in this study. For example, the study is lacking randomized control trials and a robust standardization of the intervention protocol to help control for variability in implementation to prevent bias from influencing results. Perhaps the term “effectiveness” might be more appropriate for this study rather than the term efficacy.

* The participants of this study were all from one dental school-- Baqai Dental College in Karachi, Pakistan. A multiple center study to include more schools and final-year dental students would provide a more robust data set because a larger sample size will allow for more statistical power in the data analysis methods. A sample size of 69 final-year dental students from one dental school cohort are too few to publish “efficacy” data on the mindfulness intervention examined in this study. Perhaps it might be more appropriate to frame the study as a pilot study instead.

* There should be discussion and quantitative statistical analysis to examine if (or to what extent) the demographic data collected -- such as gender -- may have influenced the study’s pre-survey and post-survey burnout findings. For example, there is evidence in the literature that female students tend to experience more stress and burnout when compared to their male counterparts.

* While the use of the validated Copenhagen burnout instrument is applauded, more description regarding the Copenhagen instrument in the analysis of quantitative data is needed. For example:

- How many survey items are in each subscale (Personal BO- 6 items?; BO related to studying- 7 items?; BO related to colleagues- 6 items?; teacher-related BO- 6 items)?

- Are survey items scored by frequency?

- What is the scoring scale for each item [“Never (0)” to “Always (4)”]?

- Is there reverse scoring for any of the survey items? If so, which items have reverse scoring?

**While the authors and this reviewer are familiar with the Copenhagen burnout instrument, the audience likely is not as familiar with the instrument. A detailed description of the instrument’s subscales, subscale items, and the scoring of items, are essential to the audiences’ understanding/interpretation the of the quantitative data.

Overall, the study presents a promising concept. However, it requires further refinement and more detailed description within the manuscript.

**Do you want your identity to be public for this peer review?** For information about this choice, including consent withdrawal, please see our Privacy Policy

Reviewer #1: No

Reviewer #2: No

---

## [Author Response · Author response to Decision Letter 1]

17 Nov 2025

Response to Reviewers

We sincerely thank the Academic Editor and both reviewers for their constructive feedback on our manuscript “Effectiveness of Mindfulness Practices in Mitigating Burnout Among Final-Year Dental Students.” We have carefully addressed each comment and revised the manuscript accordingly. Reviewer comments are presented in italics, followed by our responses.

Editorial Office Comments

Manuscript formatting and file naming

Response: Manuscript revised to comply with PLOS ONE formatting and file naming requirements. Figures and Supporting Information files now include captions.

Data availability and ethical restrictions

Response: Data contain potentially identifiable information and cannot be publicly shared due to ethical restrictions by the Baqai Medical University Ethics Review Board. Qualified researchers may request access by contacting registrar.secretariat@baqai.edu.pk

(Pages 24, Lines 631–638).

Non-author contact for data access

Response: Requests may be directed to the Baqai Ethical Review Board (IRB) at registrar.secretariat@baqai.edu.pk

, which handles all data access requests. Data are securely stored in the institutional repository.

Ethics statement location

Response: Ethics statement relocated exclusively to the Methods section.

Figure captions

Response: Added separate captions for all figures.

Supporting Information captions

Response: Captions added and in-text citations updated.

Reviewer #1 Comments

Abstract and introduction

Response: Streamlined abstract; added context about dental education in Pakistan, including program duration, daily schedules, and assessment methods (Pages 3–4, Lines 81–90).

Participant selection clarification

Response: 12 participants were stakeholders for curriculum development, not students, and were not part of the burnout assessment cohort (Pages 5–6, Lines 144–147).

CBI-SS tool selection

Response: Chosen for reliability, validity, and practicality in assessing multiple dimensions of student burnout (Page 7, Lines 194–198).

Timing between mindfulness sessions

Response: Both sessions held on the same day with a 45-minute break; noted in Limitations as a potential factor influencing outcomes (Pages 8 & 23, Lines 215–219; 592–595).

Participation rate

Response: 100% of final-year students participated; no exclusions applied. Limitations discussed (Page 6, Lines 156–160; Pages 22–23, Lines 586–588).

Wordy results/discussion language

Response: Rephrased figurative language (“weary sailors,” “provided a rich texture”) for concise academic tone (Pages 11 & 21, Lines 317–320; 540–541).

Focus group effect on colleague/teacher-related burnout

Response: Focus groups informed session content; individual-level mindfulness reduced personal/study-related burnout, but colleague/teacher burnout likely requires system-level interventions.

Limitations

Response: Detailed section added discussing single-institution sample, final-year students only, short-term data collection, and timing effects (Pages 22–23, Lines 583–602).

Burnout sensitivity

Response: Noted potential under-/over-reporting and time-of-year effects on burnout; included in Limitations (Pages 23, Lines 597–600).

Reviewer #2 Comments

Burnout as a syndrome

Response: Manuscript revised to frame burnout as burnout syndrome (BO) with background from Maslach, Jackson, and WHO (Pages 3, Lines 60–68).

Efficacy vs. effectiveness

Response: Term “effectiveness” used throughout to reflect quasi-experimental design without randomization.

Single-institution study/sample size

Response: Study aimed to assess effectiveness in Pakistan; limitations of generalizability acknowledged.

Demographic analysis

Response: Demographics collected; subgroup analyses beyond current scope. Future studies suggested.

Copenhagen Burnout Inventory details

Response: Methods section expanded with subscale, items, scoring, and reverse scoring information (Pages 7, Lines 200–207).

We thank the Academic Editor and reviewers for their constructive feedback. Manuscript revised to improve clarity, methodological justification, and alignment with journal standards. We believe the revised manuscript meets publication requirements.

---

## [Editor Report · Decision Letter 1]

19 Dec 2025

Dear Dr. Yasir,

Thank you for submitting your manuscript to PLOS ONE. After careful consideration, we feel that it has merit but does not fully meet PLOS ONE’s publication criteria as it currently stands. Therefore, we invite you to submit a revised version of the manuscript that addresses the points raised during the review process.

We look forward to receiving your revised manuscript.

Kind regards,

Jenny Wilkinson, PhD

Academic Editor

PLOS One

Journal Requirements:

**Additional Editor Comments:**

1. Throughout the manuscript p=0.000 should be replaced with p<0.001; additionally, please provide exact p values for those values >0.001

2. The number of decimal places for numerical data adjusted to match an appropriate level of precision, for example SD would normally only be reported to 2 decimal places. For most data a maximum of 2 decimals would be appropriate (except p values which have 3).

3. Please remove colour from Tables and check labels are all correct and in full (i.e. not abbreviated form that may have been used during analysis).

4. Please do not refer to differences where p>0.05 as tends, showing improvement in positive direction or indicative of differences. If the p value does not meet your pre-determined threshold then it is appropriate to make statements that imply differences.

5. Suggest moderating the language for the manuscript to avoid use of phrases such as “A bold and imperative objective …” or “… there were alarmingly high levels of exhaustion …”.

---

## [Author Response · Author response to Decision Letter 2]

12 Jan 2026

All points raised have been carefully addressed in the revised manuscript. A detailed, point-by-point response is provided in the uploaded document titled “Response to Reviewers.”

---

## [Editor Report · Decision Letter 2]

15 Jan 2026

Investigating the effectiveness of structured mindfulness sessions in mitigating burnout among final-year dental students: A mixed-methods analysis

PONE-D-25-38768R2

Dear Dr. Yasir,

We’re pleased to inform you that your manuscript has been judged scientifically suitable for publication and will be formally accepted for publication once it meets all outstanding technical requirements.

Kind regards,

Jenny Wilkinson, PhD

Academic Editor

PLOS One
---

## [Editor Report · Acceptance letter]

PONE-D-25-38768R2

PLOS One

Dear Dr. Islam,

I'm pleased to inform you that your manuscript has been deemed suitable for publication in PLOS One. Congratulations! Your manuscript is now being handed over to our production team.

Kind regards,

on behalf of

Dr Jenny Wilkinson

Academic Editor

PLOS One